# Quantum dot LEDs emitting broadband vortex beams

Guillaume Boulliard[1], Iännis Roland[1], Domitille Schanne[1], Marie Petolat, Pascal Filloux[1], Emmanuel Lhuillier [2] & Aloyse Degiron [1] ✉

The past few years have witnessed impressive developments in optical sources capable of emitting structured forms of light, such as optical vortices or vector beams. Because structured beams result from carefully engineered interferences, their synthesis requires coherent light and all the sources demonstrated so far rely on coherent lasing cavities—usually pumped with external optical schemes. Here, we introduce non-lasing sources emitting directional vortex beams upon electrical injection. Their architecture consists of colloidal PbS quantum dot LEDs that integrate a photonic environment with two complementary functions: to make the emitters populate radial photonic modes with extended spatial coherence, and to structure the leakage of these modes into free space. Our electrically-pumped sources exhibit phase singularities across the electroluminescence spectrum of the quantum dots, leading to vortex light emission with a bandwidth of 300 nm in the near-infrared.

Despite their ubiquitous use in everyday life and numerous applications in science and engineering, light-emitting diodes (LEDs) remain the object of intense developments. Chief among them is the quest to adapt these compact and efficient light sources to an ever-increasing number of electromagnetic windows, with a current push toward the deep-ultraviolet on one hand[1], and infrared wavelengths longer than 2 μm on the other hand[2,3]. In the visible and near-infrared where LEDs have already replaced almost all other non-lasing photon sources, there is no lull in the competition to investigate new materials (perovskites[4,5], semiconducting nanowires[6], colloidal quantum dots (QDs)[7,8], nanoplatelets[9], organic emitters[10,11]) that could lead to lower costs, superior color purity (for displays), higher modulation speeds (for near-infrared telecommunication wavelengths) and/or additional functionalities (e.g., flexible devices).

Regardless of the targeted application or emission range, one of the key problems that must be addressed when designing an LED is the light extraction from the active region. Even with the best materials and charge injection schemes, a significant part of the emitted light remains trapped within the devices because LEDs are made of semiconductors with high-refractive indices, inducing both partial and total internal reflection at their interfaces with the outside environment. In addition to the anti-reflection coatings, back-reflectors, and non-orthogonal facets that have long been used in commercial LEDs to improve their light extraction[12], other strategies based on maximizing the scattering and/or diffraction of light outside the devices have also been investigated. These approaches rely on integrating a structured medium within the LED stack, such as a microlens array[13,14], a diffraction grating[15,16], a photonic crystal[17], a metasurface[18,19], or an ensemble of optical antennas[20,21] or optical scatterers[22,23]. For colloidal QD LEDs that emit photons through a transparent substrate, light extraction efficiency can also be substantially improved by patterning the outer facet of the substrate with judiciously designed patterns[24,25].

Interestingly, many of these LEDs also beam light in preferential directions and/or with a high degree of linear polarization[14,16–20]. This point is particularly noteworthy because LEDs, contrary to lasers, are incoherent sources that operate in the regime of spontaneous emission, implying that each point generates photons randomly and independently from the others. Such a lack of long-range spatial correlations across the light-emitting area makes it very challenging to impart specific phase and polarization properties to the emitted light. In the examples cited above, directional and/or polarized emission occurs whenever the presence of the structured medium forces the emission of photons into well-defined optical modes within the LED stack, which in turn can only release light in free space with their own

[1]Université Paris Cité, CNRS, Laboratoire Matériaux et Phénomènes Quantiques, Paris, France. [2]Sorbonne Université, CNRS, Institut des NanoSciences de Paris, Paris, France. ✉e-mail: aloyse.degiron@u-paris.fr

polarization and along directions that obey to the laws of momentum conservation. Yet, the control over the light produced by an LED remains quite limited in comparison to the possibilities offered by lasers, which can be designed to emit very complex forms of light. Several studies have demonstrated laser cavities emitting beams with inhomogeneous phase and polarization profiles such as optical vortices, vector beams, and self-healing Airy beams[26–31]. These demonstrations are typically performed with optically-pumped cavities rather than electrically-pumped ones.

In principle, the structured medium approach can be extended to make LEDs emit beams as complex as those obtained with lasers. Recent theoretical developments indicate that full control of the spontaneous emission is possible with photonic metasurfaces that leverage local and delocalized interactions[32]. On the experimental side, advanced manipulation of the spontaneous emission has been reported with thermally- or optically-pumped samples, with the demonstration of unidirectional beams[33–35] and beams carrying phase and/or polarization singularities[36–38]. However, applying these ideas to LEDs is a non-trivial matter, as the structures demonstrated to date are not compatible with electrical pumping. The charge injection scheme of an LED represents a strong design constraint, as it must not destroy the optical interactions that are necessary to synthesize the beams.

Here, we experimentally demonstrate QD LEDs that emit directional vortex beams. Our structures are very compact (less than 500 nm thick) and generate optical vortices with a bandwidth of 300 nm in the near infrared, exploiting the broad spontaneous emission spectrum of our colloidal QDs. While broadband vortices can also be synthetized with optical elements placed outside conventional LEDs and other non-lasing sources (using spatial light modulators, spiral phase plates, volume phase holograms, uniaxial crystals, often combined with polarizing and/or spatial filtering elements)[39–42], this study shows a path forward for advanced beam structuration by directly acting upon the spontaneous emission process within the sources, facilitating their integration into more complex systems and offering potential alternatives to structured lasing light in applications where ocular safety and warmer colors are needed.

## Results

To generate vortex beams, we adapt the ideas that we have previously developed with photoluminescence experiments[36]. Our approach consisted of making colloidal quantum dots emit radial surface plasmons with extended spatial coherence at the center of a spiral hologram imprinted at the surface of an Au film. The pitch of the spiral was calculated to make the surface plasmons diffract light in a hollow cone oriented perpendicular to the surface. The topological singularity at the center of the spiral ensured that the beam was emitted with a phase singularity and helical wavefronts typical of optical vortices. To obtain these results, a key requirement was that the radial plasmons were selectively excited at the center of the structures. In photoluminescence experiments, this condition was easily fulfilled by coating a uniform layer of PbS QDs on the Au film and by selectively pumping the dots at the center of the spiral using a focused laser beam. Illuminating the full structure would not have satisfied this condition because each QD emits electromagnetic radiation independently from the other, preventing the construction of a structured phase across the extended light-emitting area[37].

We now transpose these ideas to electrically-pumped devices. As will be demonstrated later, these sources have been designed to sustain two families of guided radial modes within the LED stack. These radial waves interact with an Au spiral that out-couple them into free space in the form of an optical vortex (Methods section). The samples are schematically depicted in Fig. 1a. The first two layers of the stack are an Al cathode and a mesoporous electron transport layer made of partially sintered anatase $TiO_2$ nanocrystals. The Au spiral, with a radial pitch of 1200 nm, is then defined by electron-beam (e-beam)

lithography, metal deposition, and lift-off. An insulating spacer of soft-baked PMMA with a thickness of 250 nm is subsequently coated directly onto this spiral. We use PMMA for three reasons: (i) it has a high resistivity exceeding $2 \times 10^{15}$ $\Omega$.cm[43], (ii) it is optically transparent in this spectral range, (iii) it is a positive-tone e-beam resist, allowing us to create a 3 μm wide aperture in it aligned with the center of the spiral by e-beam lithography. The next layer is the active medium composed of n-doped PbS QDs cross-linked with mercaptocarboxylic acid (MPA) and emitting light between 1200–1600 nm. The QDs are spin-coated uniformly onto the sample, but only those filling the central aperture within the PMMA layer will experience charge injection. A hole transfer layer, made of p-doped PbS QDs cross-linked with 1,2-ethanedithiol (EDT), is then spun on the sample. This layer has a wider bandgap than the underlying emissive n-doped layer. Its emission and absorption bands occur at wavelengths smaller than those emitted by the n-doped QDs, so that its optical properties can be approximated as those of a dielectric layer with relatively few losses. Finally, the sample is covered by a transparent anode made of indium tin oxide (ITO), a material that simultaneously acts as a high-index dielectric medium at visible and near-infrared wavelengths and as a conductor of electrical current. The QD synthesis and full fabrication procedure are detailed in sections 2 and 3.1 of the Supplementary Information. It should be noted, in particular, that the spiral is slightly truncated so as to leave a 1.5 μm wide circular area without Au at the very center of the structure. Without this truncation, many n-doped QDs within the central PMMA hole would have been in contact with the Au spiral rather than with the electron transport layer, resulting in heterogeneous injection conditions that would have degraded the quality of the light emission.

Figure 1b shows a top-view image of a fabricated LED, while the top panel of Fig. 1c shows the infrared light that it produces upon electrical pumping. Clearly, light originates from the very center of the sample, validating our preferential charge injection scheme in which the light-emitting QDs are not electrically connected except at the very center of the sample. The EL spectrum, plotted on the bottom panel of Fig. 1c, features a broad peak spanning from wavelengths below 1200 nm to wavelengths above 1600 nm that reflects the distribution in size of the individual QDs.

To analyze the structure of the EL produced by our LED, we plot in Fig. 1d its experimental dispersion relation recorded with a Fourier imaging setup coupled to an imaging spectrograph (Methods section). The figure represents a map of the EL intensity under a 6 V bias voltage as a function of the radial wavevector $k_{//}$ and the wavelength $\lambda$. In addition, we have normalized $k_{//}$ by the free space wavevector $k_0 = 2\pi/\lambda$, meaning that the abscissa represents the sine of the emission angle $\theta$. Because the dispersion relation is symmetric with respect to the origin, we only plot negative values from $k_{//}/k_0 = -0.65$ (corresponding to light emission at an angle $\theta = \arcsin(0.65) = 41°$), to $k_{//}/k_0 = 0$ (corresponding to light emitted normally to the surface).

The dispersion relation features two sharp branches with opposite slopes (labeled 1a and 1b in Fig. 1d) that intersect at the origin at a wavelength $\lambda = 1475$ nm, as well as a broader third branch labeled 2 in Fig. 1d. Due to the symmetry of the system, this dispersion relation can be obtained for any azimuthal orientation of the radial wavevector $k_{//}$. This observation indicates that the light emitted by the LED at any wavelength is predominantly distributed into two directional hollow cones—one, corresponding to branches 1a or 1b, depending on whether $\lambda$ is smaller or larger than 1475 nm, and another one, corresponding to the broader branch 2.

To understand the origin of these branches, we have performed full-wave simulations with a commercial finite element code. Rather than modeling the full LED stack, we have exploited the fact that the Au spiral resembles a linear periodic grating away from its center. This allowed us to approximate the structure as a two-dimensional grating and to compute its reflectivity in a single unit cell flanked by periodic boundary conditions. As in our past simulations of functional LEDs[20,44],

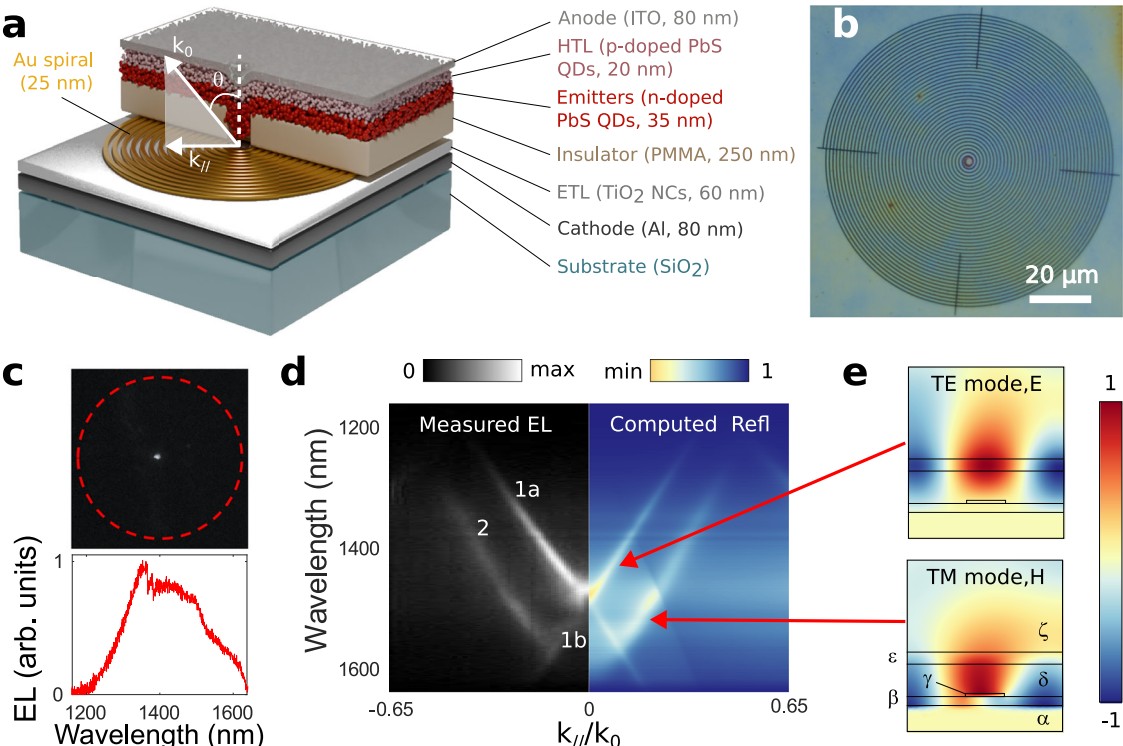

**Fig. 1 | QD LEDs operating with Au spirals. a** Schematic drawing of the LED, with half of the upper layers not represented so as to visualize the Au pattern. HTL and ETL stand for hole transfer layer and electron transfer layer, respectively, while NCs stands for nanocrystals. **b** Top view image of a fabricated device. The four straight bars that are visible have been added to improve the structural integrity of the e-beam resist after development of the spiraling pattern. **c** Near-infrared image of the EL emitted by the device (top) and measured EL spectrum (bottom). The red dashed circle added to the top panel marks the outer perimeter of the Au spiral. **d** Experimental dispersion relation of the EL at 6 V (negative values of $k_{//}/k_0$) compared to the computed reflection of a two-dimensional linear grating that approximates the behavior of the real structure (positive values of $k_{//}/k_0$). Labels 1a, 1b, and 2 on the experimental map identify the different branches discussed in the text. **e** Computed instantaneous electric field of the TE mode at $\lambda = 1407$ nm and $k_{//}/k_0 = 0.1$, and computed instantaneous magnetic field of the TM mode at $\lambda = 1520$ nm and $k_{//}/k_0 = 0.16$ supported by the two-dimensional linear grating. The Greek letters on the bottom map identify the different layers of the model: α: Al, β: TiO$_2$, γ: Au, δ: PMMA, ε: effective layer merging the n-doped QDs, p-doped QDs, and ITO, ζ: air.

we do not explicitly include the thin QD layers in our model but instead merge them into a single effective dielectric layer together with the high-index ITO anode (Supplementary Information, section 1). The results of these calculations, which are plotted for positive values of $k_{//}/k_0$ in Fig. 1d, almost perfectly mirror the dispersion relation of the experimental EL. According to these simulations, the branches labeled 1a and 1b that cross at $\lambda = 1475$ nm correspond to the excitation of a dielectric (i.e., non-plasmonic) TE-polarized mode guided in the top layers of the device, as shown with the electric field map of Fig. 1e. The spatial coherence length of this mode is given by $\lambda/\Delta\theta$, where $\Delta\theta$ is the branch width in radians. Using the experimental results, we find a coherence length of 34 μm for branches 1a and 1b.

In contrast, the dispersive branch 2 corresponds to the excitation of a TM-polarized mode. An examination of its magnetic field distribution, displayed in the lower panel of Fig. 1e, reveals that this mode is a hybrid dielectric-plasmonic wave guided between the Al electrode and the PMMA layer. More specifically, the guiding mechanism does not only involve a coupling with the surface plasmons of the Al/TiO$_2$ interface, but also a hybridization with the surface plasmons of the Au metallic grating (see also Supplementary Fig. 4 for the electric field distribution of these hybrid waves). These plasmonic contributions, involving metals with sizeable material losses, explain why this TM mode manifests itself as a broader branch in the dispersion relations of Fig. 1d. Despite this broadening, the spatial coherence length $\lambda/\Delta\theta$ of this mode reaches 16 μm, which remains many times larger than the radial periodicity of the spiral.

The two families of modes evidenced by our simulations are reminiscent of those recently observed for III-V QDs interacting with

linear Au gratings coated with a 285 nm thick layer of PMMA[45]. However, in the present case, the experimental branches are not pure TE- or TM-polarized modes but rather radial waves generated by the QDs at the very center of the spiral and subsequently diffracted as directional hollow cones in free space by the spiral. Since the EL signal in Fig. 1d does not fall back to strictly zero outside the TE- and TM-like branches, this directional emission is accompanied by a fainter isotropic background corresponding to the fraction of the EL that is not coupled to the modes of interest.

We now show that the EL possesses a broadband phase singularity. A powerful method to characterize the phase of a beam consists in making it successively interfere with four waves with planar wavefronts incrementally shifted by π/2[46,47]. From the four resulting patterns $I_1$, $I_2$, $I_3$ and $I_4$, the experimental phase of the beam under investigation can be retrieved by evaluating $\arctan\left[(I_4 - I_2)/(I_1 - I_3)\right]$. A difficulty in our case is that LEDs emit photons that have no temporal coherence, making it impossible to probe them with an external reference beam. We overcome this problem by fabricating a series of four interferometric LEDs that superimpose the Au spiral of the sample examined so far with Au bullseye gratings that emit planar wavefronts with the desired phase shifts, as schematically depicted in Fig. 2a. The radial pitch of these bullseye gratings is 1200 nm just as that of the spiral, forcing the light emission into the same wavelength-dependent hollow cones as those evidenced in Fig. 1d. Scanning electron micrographs of the resulting interferometric structures are displayed in Fig. 2b and Supplementary Fig. 2, while the corresponding EL emission patterns of the four fabricated LEDs are represented in Fig. 2c. These emission patterns have been obtained by visualizing the EL emission of the four

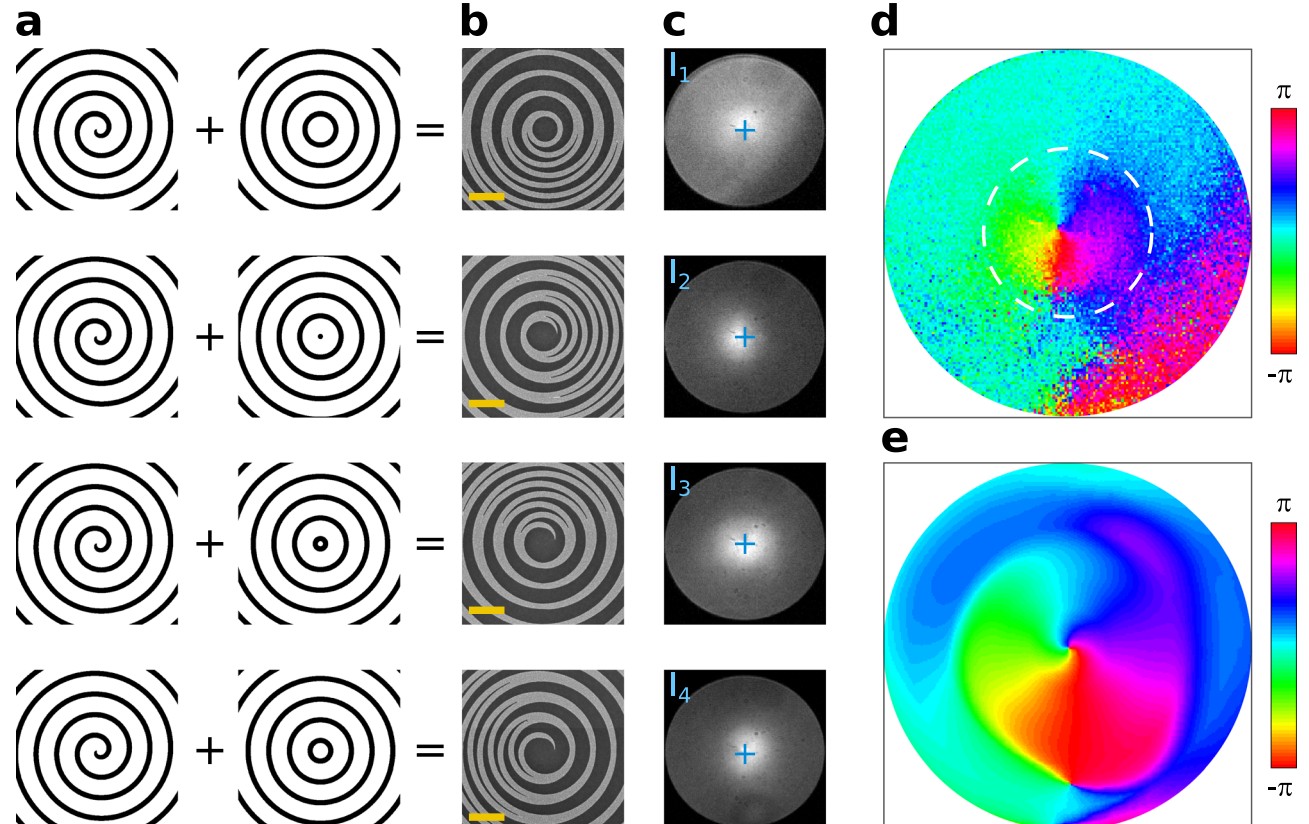

**Fig. 2 | Experimental demonstration of the phase singularity. a** Design principle of the four interferometric LEDs, which each feature the same Au spiral superimposed with a different bullseye grating. All bullseye gratings have the same periodicity but a different phase at the origin. **b** Scanning electron micrograph of the resulting Au patterns. The yellow scale bar represents 2 µm. Note that the very center of each pattern has been intentionally omitted in the fabrication process to ensure proper electrical injection. **c** Back focal plane images of the EL emitted by

each interferometric LED. The blue cross represents the center of the images. All the information is contained in a circle with a radius $k_{//}/k_0 = 0.65$, corresponding to the numerical aperture of our microscope objective. **d** Experimental phase deduced from the four interferometric patterns shown in panel c. The white dashed circle is a guide for the eyes with a radius $k_{//}/k_0 = 0.3$. **e** Calculated phase obtained with an analytical model assuming that the LEDs operate with scalar radial waves launched at the very center of the spirals.

LEDs in the Fourier space, which is equivalent to examine the EL on a screen placed far away from the devices. Importantly, each of these four images contain all the emitted wavelengths, from below 1200 nm to above 1600 nm as previously measured in Fig. 1c. They show that light is predominantly emitted slightly away from the optical axis, visualized by a blue cross in the panels of Fig. 2c, and in a different direction for each of the four LEDs. By plugging these four experimental images in the argument of the arctan function discussed above and after unwrapping the phase between -π and +π using basic trigonometry[46], we finally retrieve the experimental phase distribution of the beam emitted by QD LEDs featuring a spiral grating. This phase, plotted in Fig. 2d, clearly rotates from -π to +π around a central singularity. This pattern unambiguously demonstrates that the LED presented in Fig. 1 is generating a vortex beam.

This result is a direct consequence of the efficient coupling between the QDs that are electrically pumped at the center of the structure and the two families of guided radial modes discussed with Fig. 1d. Because these modes with extended spatial coherence are diffracted by a spiraling Au grating with a topological singularity at its center, the out-coupled beam acquires the experimental phase profile evidenced in Fig. 2d. To support this interpretation, we have developed a scalar analytical model that includes these key elements and computes the far-field patterns of the interferometric LEDs in the Fraunhofer approximation. This model, described in the Methods section, successfully reproduces the phase singularity observed in the experiments, as can be appreciated in Fig. 2e. The agreement between measurements and calculations is best within the dashed circle that we

have superimposed on the experimental data of Fig. 2d. This circle, with a radius $k_{//}/k_0 = 0.3$, corresponds to the in-plane wavevector value where all the experimental branches of the dispersion relation of Fig. 1d have disappeared in the faint isotropic background, at λ ≈ 1280 nm. For shorter wavelengths and larger wavevectors, the measurements of Figs. 1d and 2d are dominated by the isotropic background and the vortex emission becomes negligible, explaining why discrepancies between measurements and theory arise outside the white dashed circle of Fig. 2d. As a corollary, the phase singularity is unambiguously encoded across the spectral range where the TE- and TM-like branches of Fig. 1d stand out from the noise, from λ ≈ 1280 nm to λ ≈ 1580 nm. More precisely, since the modes of interest are emitted in a hollow cone with a wavelength-dependent aperture (Fig. 1d), the broadband nature of the vortex emission as well as the contribution of the different wavelengths to the experimental phase pattern are directly visible in Figs. 2d and 2e, with the wavelengths closest to the crossing of the TE-like branches 1a and 1b contributing to the signal at the center of the image and the other wavelengths contributing to the signal at increasingly longer distances from this center (see also Supplementary Fig. 6 for a more detailed analysis).

The results of Fig. 2 demonstrate that the stringent requirements for non-trivial beam synthesis and those for electrical pumping can be simultaneously met in an incoherent light source. A key element in this success is the PMMA spacer that simultaneously acts as a mask for selective electrical injection at the center of the structures and as an optical medium in which guided modes with extended spatial coherence develop. We observed, however, that the fabricated LEDs

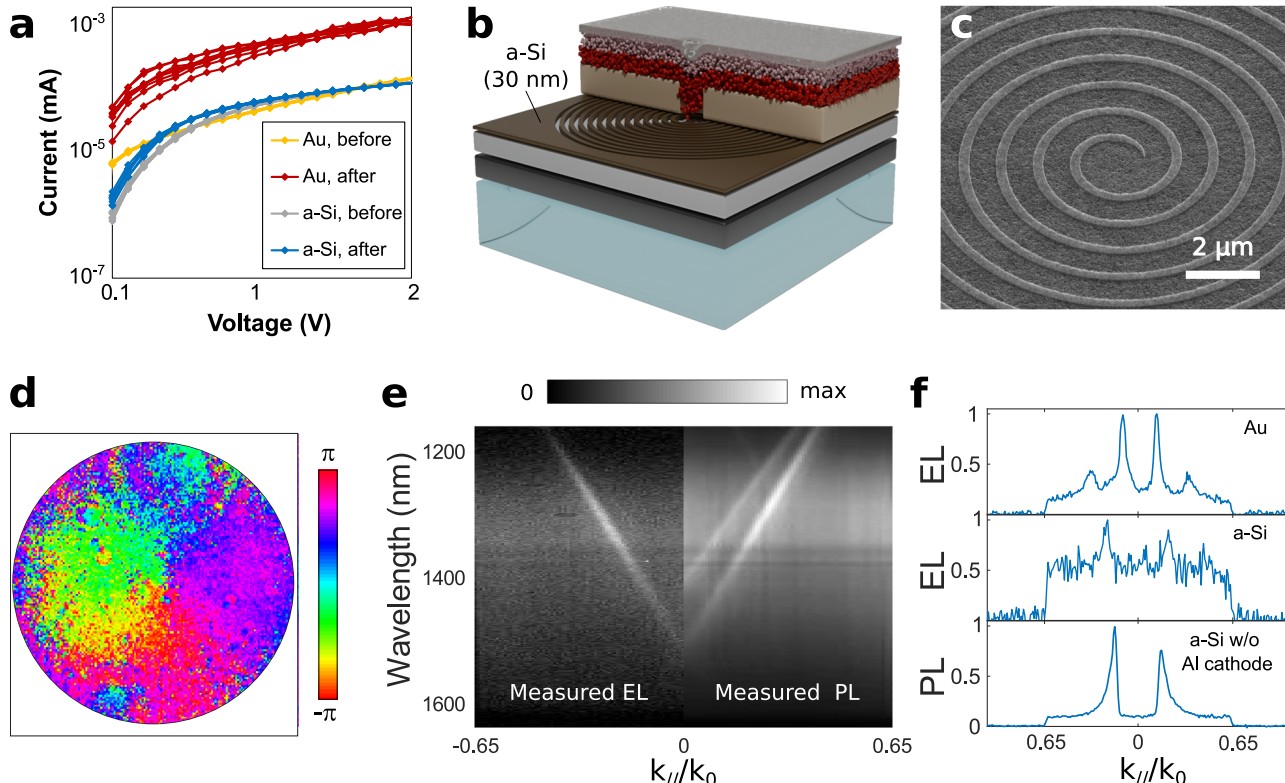

**Fig. 3 | Improving the robustness of electrical pumping. a** Current-voltage characteristics of the LEDs presented in this study. Each group of curves represents four cycles from 0 V to 2 V and back to 0 V. Yellow: LED presented in Fig. 1, before extensive characterization. Red: LED presented in Fig. 1 shortly before failure. Gray: LED presented in Fig. 3, before extensive characterization. Blue: same as yellow, but after extensive characterization. **b** Schematic of the LED operating with an a-Si spiral. The different layers are the same as before, except that Au has been replaced by a-Si. **c** Scanning electron microscope image of an a-Si spiral on top of the $TiO_2$ electron injection layer (tilted view). As before, the very central part of the spiral is omitted to allow for proper electrical injection. **d** Experimental phase of the EL measured with four interferometric devices. **e** Experimental EL dispersion relation

at 10 V (negative values of $k_{//}/k_0$) compared to the experimental dispersion relation in photoluminescence (positive values of $k_{//}/k_0$). The photoluminescence data have been obtained by pumping the center of the device with a focused laser spot. **f** Cross-section of the EL dispersion relations for the LEDs operating with an Au spiral (top) and a-Si (middle). The bottom panel represents a cross-section of the dispersion relation of the photoluminescence emitted by a device operating with a-Si, but without the bottom electrode (the full characterization of this device is plotted in Supplementary Fig. 8). These cross-sections have been evaluated at wavelengths close to the maximum of luminescence (1405 nm for the Au pattern and 1330 nm in the case of a-Si).

typically cease to work after a variable number of experiments, and especially if they are subjected to voltages larger than 6 V. The failure of a working LED is associated with a dramatic change in its electrical characteristics, the current jumping by one order of magnitude, as shown with the yellow and red curves of Fig. 3a for the LED previously characterized in Fig. 1. The relative fragility of the devices can be unambiguously traced to the Au spiral, which seem to induce heterogeneities in the applied electrical current and equipotential lines that are substantial enough to induce irreversible shorts through the PMMA layer. When fabricating control samples without Au patterns within the LED stack, all the LEDs light up upon electrical injection, and the fabricated devices exhibit high robustness upon prolonged electrical pumping. These observations indicate that replacing PMMA by a better insulator will enhance the robustness of the devices. Alternatively, replacing the Au spiral with a non-conducting dielectric one should lead to similar electrical improvements. In this case, the replacement material must also have a refractive index that provides sufficient contrast with the other layers of the stack to have an impact on the modes guided within the LED. The remaining of this study explores this avenue using 30 nm thick amorphous silicon (a-Si) patterns as replacements of the Au patterns.

A schematic of the new structure is shown in Fig. 3b. The fabrication workflow remains the same as before, except for the fabrication of the a-Si spiral that is obtained with a combination of plasma-enhanced chemical vapor deposition, e-beam lithography and reactive

ion etching (Fig. 3c and section 3.2 of the Supplementary Information). The rationale behind the choice of a-Si is that it is an insulating material that should not generate an electrical short as Au does. At the same time, it has a much higher refractive index than the other neighboring layers of the stack, which should lead to efficient out-coupling of the modes guided within the device into free space (we note in passing that the radial pitch of the spiral has been tuned to 1000 nm to take into account the refractive index change compared to Au). Intensity-voltage curves, measured before and after extensive EL characterization over the course of two weeks, sometimes at voltages exceeding 10 V, prove the robustness of this new design: the two measurements are almost the same (gray and blue curves of Fig. 3a) and also resembles the characteristic of the LED operating with the Au spiral prior to its failure (yellow curve of Fig. 3a).

To verify that LEDs operating with a-Si spirals generate broadband optical vortices, we have fabricated four interferometric LEDs following the strategy already outlined in Fig. 2, from which we have derived the phase profile of the beam. The resulting map, shown in Fig. 3d, features the expected phase singularity typical of a vortex beam with topological charge $m = 1$ at its center. The measurement, however, is noisier than in the case of the LED operating with an Au spiral for reasons that will become apparent later.

To pursue the characterization, we plot in Fig. 3e the EL dispersion relation of a non-interferometric LED featuring an a-Si spiral. Unlike LEDs operating with Au spirals (Fig. 1d), only one dispersive branch is

visible in the dispersion map, with a linewidth corresponding to a spatial coherence length $\lambda/\Delta\theta$ of approximately 25 μm. Additional polarization measurements, plotted in Supplementary Fig. 7, reveal that this branch is a hybrid plasmon-dielectric mode with a local TM polarization along any propagation radius, similar to branch 2 in Fig. 1d. By contrast, the dispersion relation of the same structure, but this time optically pumped with a laser spot at the center of the a-Si spiral, features an additional branch that is the equivalent of the TE-like branch 1a in Fig. 1d according to polarization measurements plotted in Supplementary Fig. 7.

The photoluminescence experiments of Fig. 3e and Supplementary Fig. 7 demonstrate that the modes supported by an LED with an a-Si spiraling grating are similar to those evidenced in Fig. 1 for an LED operating with an Au spiral. The fact that only one of these branches is excited in EL (left side of Fig. 3e) while two branches are excited in photoluminescence (right side of Fig. 3e) can be understood by the difference in the pumping area. In EL, only the QDs filling the aperture in the PMMA spacer are electrically connected. In the photoluminescence experiments, the size of the laser excitation spot is slightly larger than the aperture in the PMMA spacer, suggesting that the top branch of the dispersion relation is predominantly fed by the QDs at the periphery of the aperture in the PMMA spacer. In any case, the isotropic background evidenced for both EL and photoluminescence experiments indicates that part of the light exits the structure without being coupled to the guided modes involved in the vortex beam generation. This background emission is higher than for LEDs operating with Au spirals, as can be appreciated with the top two curves of Fig. 3f representing a cross-section of the EL dispersion relation of both structures at a wavelength close to the maximum emission, and thus explains why the phase singularity measurement of Fig. 3d is noisier than in Fig. 2d.

To understand the origin of this background, it is useful to study the photoluminescence of a similar stack, but without the bottom Al cathode. As shown with the experimental dispersion relation of Supplementary Fig. 8a, almost all the photoluminescence is distributed in two well-defined crossing branches, with minimal signal outside these branches. In fact, the level of isotropic noise is the lowest of all the samples presented in this study, as shown in the bottom panel of Fig. 3f, representing a cross-section of this dispersion relation. These measurements imply that the isotropic background of the LED operating with an a-Si spiral is caused by the Al electrode. Additional full-wave simulations, presented in Supplementary Fig. 8b, explain this behavior by the fact that the guided modes of a stack without a bottom Al cathode have a significant field extension within the glass substrate beneath the a-Si spiral. When an Al cathode is present, the field cannot extend into the substrate anymore, altering the interactions with the QDs as well as the outcoupling of the guided modes into free space. Nevertheless, these results demonstrate the potential of a-Si photonic structures for molding the light emission in very complex ways (in this regard, it is also worth noting that the photoluminescence experiments of Supplementary Fig. 8 show that the device without an Al cathode generates a vector beam with an azimuthal polarization texture).

## Discussion

We have demonstrated compact QD LEDs that emit directional broadband vortex beams with unity topological charge. By so doing, we have shown, with two different architectures, how to conciliate the seemingly incompatible conditions for advanced control of the phenomenon of spontaneous emission and for electrical pumping. The same approach can be readily adapted to fabricate LEDs that emit vortex beams with higher topological charges, using spiral holograms with more than one branch. While the results presented in this study rely on restricting the diameter of the pumping area to approximately one wavelength, it is not necessarily a limitation. In particular, this

approach shows promise to reach the regime of single-photon emission if one replaces the continuous layer of PbS colloidal QDs with a single emitter at the center of the stack. Further work is required to obtain the same behavior without restricting electrical pumping to the center of the structure, which is possible in theory, although the current proposals are not readily compatible with electrical injection[32]. In any case, the present study contributes to enriching the array of functionalities that LEDs can offer, with features that are usually associated with lasers and in a compact format that is favorable for potential applications, from wireless optical communications that benefit from the extended bandwidth of transfer protocols based on optical vortices[48] to lab-on-a-chip platforms where small vortices may serve as optical probes or tweezers.

## Methods
### Design of the Au or a-Si spiraling patterns
The core idea is that the colloidal QDs that are pumped at the center of the devices can be considered as a local source of cylindrical (radial) guided waves. Two types of guided cylindrical waves with different polarization properties are supported within our LED stacks: a dielectric mode that mainly develops in the upper layers above the Au or a-Si spiral, and a hybrid mode that involves a surface plasmon bound to the bottom Al cathode. Our goal is to uncouple these guided waves regardless of their polarization and weave them into a vortex beam with a topological charge m = 1. Moreover, we restrict the electrical pumping to the center of the structure. For this, we calculate the hologram resulting from the interference between a scalar radial wave centered at the origin and an exit wave at normal incidence carrying a first order phase singularity $\exp(i\Phi)$:

$$H(x, y, \Phi) = \left| \exp\left[iRe\left(n_{eff}\frac{2\pi}{\lambda_c}\sqrt{x^2+y^2}\right)\right] + \exp\left[i\Phi + i\alpha_0\right] \right|^2, \quad (1)$$

where $n_{eff}$ is the effective index of the radial wave, $\alpha_O$ is a constant phase factor that sets the orientation of the spiral ($\alpha_O$ is arbitrarily set at $3\pi/2$ for all the masks of the study) and $\lambda_c$ is the wavelength at which the beam is emitted at normal incidence, i.e., the wavelength at which the in-plane wavevector $k_{//}$ is equal to zero in the dispersion relation. We chose this wavelength to be roughly in the middle of the emission band of our QDs. We then convert this spiraling pattern with sinusoidal variations into a binary hologram with square variations and a duty cycle of 0.25. After removing the very center of the spiral so as to avoid perturbing the pumping of the PbS QDs, this mask is finally transferred to the sample by electron beam lithography.

The holograms of the interferometric LEDs used for the phase characterization (Figs. 2b–d and 3d of the main text) are a superposition of the binary spiraling pattern derived from Eq. (1) and a binary bullseye structure. The bullseye of each of the four holograms has a different phase $\varphi_O$ at the origin, respectively set to $\varphi_O = 0$, $\pi/2$, $\pi$, and $3\pi/2$. These bullseye patterns are calculated by making a scalar radial wave centered at the origin interfere with an exit wave at normal incidence without phase singularity [this amounts at replacing the second exponential term of Eq. (1) by $\exp(i\alpha_O + i\varphi_O)$].

### A scalar analytical model that supports the phase experiments
To obtain the phase pattern of Fig. 2e, we model the guided modes supported by the LED stack as cylindrical waves emitted at the very center of the structures:

$$A_j(x, y, \lambda) \approx \exp\left[in_j k_0 \sqrt{x^2+y^2}\right] / (x^2+y^2)^{1/4}, \quad (2)$$

where the origin of the $(x,y)$ coordinates is at the center of the spiral, $k_O = 2\pi/\lambda$ is the free space wavevector and $n_j$ is the effective index of branch $j$ of the dispersion relation. In this scalar model, the vector field distribution characterizing each family of modes is not taken into

account. The real part of $n_j$ is extracted from the experimental dispersion relations by examining the wavelength at which the branch of interest crosses the origin. According to standard diffraction theory (in-plane momentum conservation), this wavelength is equal to the product of the grating periodicity by Re($n_j$). As for the imaginary part of $n_j$, it is related to the characteristic propagation length $L$ at which the intensity of the mode has decreased by 1/2e: $L = 1/[2\text{Im}(n_j)k_O]$. In our case, this propagation length is also the spatial coherence length of the mode, which can be derived from the width of the branch of interest in the experimental dispersion relation: $L = \lambda/\Delta\theta$, where $\Delta\theta$ is the branch width in radians.

At each wavelength, two radial waves exists simultaneously (but do not mutually interact due to their different polarization properties): a radial wave $A_1$, corresponding to the TE-like branch 1a or 1b, and a radial wave $A_2$, corresponding to the TM-like branch 2. From the experimental dispersion relation and the explanations of the previous paragraph, these waves have an effective index $n_1 = 1.23 + 0.003i$ and $n_2 = 1.39 + 0.007i$, respectively.

To reproduce the experimental phase pattern of Fig. 2d, we make the scalar waves interact successively with the four interferometric masks $H_1$, $H_2$, $H_3$ and $H_4$ represented in Fig. 2b and compute the resulting far field diffraction patterns in the Fraunhofer approximation. Contrarily to the actual devices, we do not remove the very center of the spiral. For each mask, we then sum the result over all the wavelengths weighted by the measured EL intensity $S(\lambda)$:

$$I_i\left(k_x, k_y\right) = \sum_\lambda S(\lambda)\left|\mathcal{F}\{H_i(x,y)A_1(x,y,\lambda)\}\right|^2 + \sum_\lambda S(\lambda)\left|\mathcal{F}\{H_i(x,y)A_2(x,y,\lambda)\}\right|^2$$

(3)

In this expression, $\mathcal{F}$ denotes a Fourier transform, while $k_x$ and $k_y$ are the projections of the emitted wavevector along the plane of the sample. From these four interferometric patterns, we finally construct Fig. 2e by evaluating $\arctan\left[(I_4 - I_2)/(I_1 - I_3)\right]$ and by converting the modulo π result to a modulo 2π map with straightforward trigonometric considerations. The agreement between experiments and simulations can be further refined by taking into account that the experimental $I_1$ pattern has a slightly stronger background compared to the other images, affecting the evaluation of the phase far from the region of the central singularity because the useful signal is weak far from the optical axis (Supplementary Fig. 9). In Fig. 2e, the phase evaluation has been performed after adding a background constant to each pixel of the computed $I_1$ image equal to 0.01% of the maximum pixel intensity of this $I_1$ image.

### Numerical simulations
Away from the center of the samples, the spirals can be approximated as linear gratings along any azimuthal direction. Therefore, we perform our full-wave simulations in two dimensions, by simulating a periodic unit cell flanked by periodic boundary conditions using commercial code (COMSOL Multiphysics, RF module). All the details are given in Section 1 of the Supplementary Information.

### Sample fabrication
All the details regarding the synthesis of the colloidal QDs, as well as the many fabrication steps that involve optical lithography and two rounds of e-beam lithography, are detailed in sections 2 and 3 of the Supplementary Information. Cross-sections of a fabricated device are shown in Supplementary Fig. 2.

### Experimental setup
All experiments were performed under an Olympus BX51WI optical microscope coupled to an NIRvana InGaAs camera (Princeton Instruments) mounted at the exit port of an Acton SP-2356 imaging spectrometer (Princeton Instruments). Electrical pumping and electrical

characterization were performed with a Keithley 2636B sourcemeter. Optical pumping for the photoluminescence (PL) experiments was performed with a HeNe laser at 633 nm focused with a LCPLN50XIR microscope objective (magnification 50X, numerical aperture NA = 0.65) at the center of the structure under investigation. Regardless of the pumping method (EL or PL), the signal is collected by the 50X microscope objective and focused to form an intermediate image at the entrance of the imaging spectrometer using a 20 mm tube lens (in the case of PL experiments, a Thorlabs DMLP950R dichroic mirror is inserted in the optical path to separate the 633 nm laser pump from the PL signal). Dispersion relations in PL or EL are obtained by inserting a 30 mm Bertrand lens behind the microscope objective so as to form an intermediate image in the Fourier space at the entrance of the spectrometer. This image is spatially filtered by a 200 μm wide slit before being dispersed along one dimension by an 85-groove/mm grating within the spectrometer. The back focal plane images of Supplementary Fig. 8 were obtained by filtering the signal with a bandpass filter centered around 1250 nm, and by replacing the 85-groove/mm grating with a planar mirror inside the spectrometer. All polarization-dependent measurements were performed by analyzing the signal with a linear wire grid polarizer (WP25M-UB from Thorlabs).

## Data availability
The authors declare that the data supporting the findings of this study are available within the paper, its supplementary information file, and the accompanying source data file for all line graphs and point plots. Source data are provided with this paper.

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

## Acknowledgements

We acknowledge support from the European Research Council Grants FORWARD (reference: 771688, A.D.) and AQDtive (reference: 101086358, E.L.). This work was also supported by French state funds managed by the Agence Nationale de la Recherche through the grant Bright (ANR-21-CE24-0012, E.L.). We acknowledge the use of clean-room facilities from the "Centrale de Proximité Paris-Centre" and support from Renatech+ for micro and nanofabrication. We would like to thank Eric Aït-Yahiatène (Laboratoire PASTEUR, Département de Chimie Ecole Normale Supérieure, PSL University - Sorbonne Université - CNRS) for assistance with the ITO deposition onto the samples.

## Author contributions

G.B. and M.P. fabricated and characterized all the devices presented in this work. D.S., I.R., and G.B., assisted by P.F., developed the architecture and fabrication workflow of the vortex-emitting LEDs. E.L. synthesized the n-doped and p-doped PbS QDs and provided guidance for their processing and for the electrical injection scheme. G.B., M.P., and A.D. ran the full-wave simulations. D.S. and A.D. developed the scalar model. M.P. made the three-dimensional drawings of Figs. 1a and 3b. G.B. and A.D. wrote the manuscript, with feedback from all authors. A.D. supervised the work.

## Competing interests

The authors declare no competing interests.
