## [Transparent Peer Review file · Nature Communications]

Quantum Dot LEDs Emitting Broadband Vortex Beams

Corresponding Author: Dr Aloyse Degiron

Version 0:

Reviewer comments:

Reviewer #1

(Remarks to the Author)

In this work the authors create a phase structured output from an incoherent source by careful design to allow electrical excitation in a small central region. Most of the figures show direct data pertaining to the device and its design, while the optical data is somewhat less convincing. The actual phase structure is shown in only one plot, Fig 2d, and then repeated later when the device is made more reliable. This is necessarily an indirect reconstruction, but that too makes the single result less appealing. Overall while the paper was easy to read, I feel that the motivation for the work is poor - why do we even need an LED emitting vortex beam when we have so many coherent sources of OAM? This critical question is never answered. What would I do with this LED device? Never mentioned. Mention of the record breaking 400 nm is said many times, but never why this is useful or desired. Broadband OAM control from white light sources has been shown in MANY paper, none of them cited. In fact, the introduction is not framed at all in terms of OAM/vortex sources, nor incoherent/broadband OAM control, with none of the literature in these two directions covered (only a modest mention in passing of a couple of papers that appear to have been selected at random). Instead it is centred on LEDs and the advances in this field. Without context or motivation, the work then comes across as a technical advance in device fabrication, with plausible confirmation. Here too though, the authors need to appreciate that there is a difference between OAM and phase singularities (see many papers on this by Michael Berry), so a swirling phase does not imply OAM. One would like to see the entire complex field to be sure we have a mode here, and for that matter, a mode of what type? I suggest that this be transferred to a journal that is read by a more specialised audience interested in solid state devices.

Reviewer #2

(Remarks to the Author)

Title: Quantum Dot LEDs Emitting Broadband Vortex Beams

Authors: G. Boulliard, I. Roland, D. Schanne, M. Petolat, P. Filloux, E. Lhuillier, A. Degiron

The authors report on an electrically pumped optical source that emits a directional broadband vortex beam upon electrical injection. The approach consists of a metallic diffractive element that is disposed beneath an active layer region of embedded colloidal quantum dots. An aperture in the interposed dielectric spacer allows local injection of carriers, thereby confining the emission to a micro-sized region at the centre of the diffractive element. The dispersion relation is retrieved by means of a spectrally-resolved Fourier imaging system and compared to the computed reflectivity of a simplified structure (a 2D grating) which approximates the system. The two are found to be in very convincing agreement. Then, in order to evidence a phase singularity over the entire range of wavelengths, a powerful method is employed. It involves the superimposition of four distinct gratings, each possessing specific phase at origin, on separate devices. The four resulting emission patterns are then combined to reveal the singularity. A vortex beam with unity topological charge is demonstrated. The weakness of the device (and its robustness in the absence of the metallic grating) when current passes through the device, leads the authors to consider a silicon diffractive element instead of metallic one. The same experimental procedure is pursued on this new architecture. Despite noisier measurements, the occurrence of singularity is once again observed, whilst robustness improved.

The article is well structured and the approach relevant. The scientific approach is rigorous. It is well written and informative. I suggest that greater emphasis be placed on the practical implications for applications of the achievements reported here. The article would be significantly enhanced in terms of its impact and the emphasis of the results would be strengthened if the authors provided a more detailed explanation of how the emission of a vortex by an incoherent source with restricted spatial coherence represents a major advance.

In light of the aforementioned considerations, it is my recommendation that this paper be considered for publication in Nature Communications, pending the implementation of any requested corrections by the reviewers.

Questions/Comments:

To my opinion, the statement "hologram" appears somewhat exaggerated in the context of a diffractive element that, in essence, constitutes a simple canonical geometry structure: namely, a spiral.

In the initial configuration, authors emphasise that no metallic pattern is introduced at the very centre of the device to allow effective electrical injection. The rationale behind the exclusion of metal, given its potential to facilitate the transport of carriers, is not immediately apparent.

The density of n-doped quantum dots in the active zone is not specified. How many active QDs are involved when the pumping zone is restricted to the 3- μm size aperture in the insulator?

A PMMA spacer is inserted between the active region and the diffractive element. What considerations led to the decision regarding the layer thickness?

As illustrated in Figure 3-e, the authors present the experimental and computed dispersion relations for Silicon-based sources. In this case, what is the coherence length?

The reasons for the significantly higher noise levels observed with silicon spirals compared to gold ones remain ambiguous. Is this discrepancy merely attributable to the lower index of refraction of silicon, or do other factors play a more substantial role?

Minor comments:

Some of the acronyms are not defined. For instance ETL. I guess this is electron transport layer. Same for HTL and NC.

Reviewer #3

(Remarks to the Author)

The authors claim to demonstrate electrically-pumped LEDs that emit vortex beams. I think the central result is of high significance and the paper combines interesting and clever fabrication, measurement, and theory/simulation. I recommend the paper for publication in Nature Communications. My main reservation are certain areas of analysis/description that I think are some combination of oversold or not clearly explained. I am happy to classify these as "minor revisions" the main structure and presentation of the paper is solid and there is no need for any "major" change.

1. In the introductory section (page 3), the authors introduce some references on optically pumped beams and then state "However, applying these ideas to LEDs is a non-trivial matter, as the structures demonstrated up to date are not compatible with electrical pumping." I think the first part of this sentence is certainly correct; the second part however misses work on electroluminescent devices.

Namely authors show unidirectional beams within electroluminescent devices in:

- "Metasurface electrode light emitting diodes with planar light control" (<https://doi.org/10.1038/s41598-017-15254-3>) and
- "Metasurface Light-Emitting Diodes with Directional and Focused Emission" (<https://doi.org/10.1021/acs.nanolett.3c03272>)

2. On page 8 the authors write that "These plasmonic contributions, involving metals with...explain why this TM mode manifests..." I don't think this is so evident from plotting the H-field intensity of the TM mode. I understand that they plot the H-field because it is characteristic of the mode, but it is the E-field intensity that determines the relative coupling to the emitter vs. the plasmonic metals. It would be nice to see this mode profile instead, or added to the supplement. In part, I wonder if the issue is more that the the TM mode couples weakly to the emitter rather than metal losses.

3. On page 10, the authors write that "This pattern unambiguously demonstrate that the LED...is generating a broadband vortex beam with a topological charge $m=1$...demonstrating that the phase singularity is encoded across the entire EL spectrum". I think they overstate their case here, and elide issues with their results that are worth discussing.

- It is my understanding that an "ideal" or "unambiguous" vortex beam with charge $m=1$ would show the 2π azimuthal phase variation at all radial distances in Fig. 2d. Instead this is only really seen in the center, and the radial distance over which the phase evolution is clearly visible is noticeably smaller than in Fig. 3e. I think it would be better if the paper was clearer and more honest about the nature of the results so far. Namely they are probably seeing some vortex beam emission superimposed on non-trivial background emission that is not strongly mediated by the spiral structures.

- I further suspect that the degree to which this looks like an "ideal" or "unambiguous" vortex beam is probably significantly wavelength dependent. It is certainly possible that at some parts of the spectrum there is no vortex beam effects evident at all. That all wavelength photons are collected to produce Fig. 2d does not prove that all wavelengths have an emission characterized by the pattern in Fig. 2d. If the authors want to show this clearly, they could use three notch filters at ~ 1200 , 1400 , and 1600 to plot something like Fig. 2d at different portions of the emission spectrum. Absent a measurement like that, I think they should be less certain and more precise about the degree to which "the phase singularity is encoded across the entire EL spectrum". Particularly in light of my previous bullet.

4. I did not understand the argument about how the Au spiral "seems to favor irreversible current leaks". I get it in the context of their device results, but I don't understand it in the context of how the device should work. Like, if I just made a planar (i.e. unpatterned) device where Au is blanket deposited atop the ETL, why would that cause "current leaks"? That whole bottom section is still the Cathode and should be electrically isolated from the anode except through current paths that pass through the semiconductor active area. My understanding is that we want to avoid direct metal contact to the semiconductor active area to eliminate recombination effects and to create selective contacts, not to prevent "current leaks". Is this just an issue with the quality of the PMMA film deposited atop Au spiral vs. when it's deposited atop a-Si? If I'm missing something, some clarifying cartoons and/or device schematics would be helpful.

Version 1:

Reviewer comments:

Reviewer #1

(Remarks to the Author)

The authors have given a much better motivation in the response letter, and somewhat better in the introduction too. With a better context I can appreciate the importance of the work to a broader audience. As for the comment that the authors are not aware of a $0-2\pi$ phase sweep in the azimuthal that does not contain OAM, a very common one would be speckle - the dark spots are phase singularities of this type but this optical field does not carry OAM.

Reviewer #2

(Remarks to the Author)

The authors have provided responses to the various questions and comments raised by the reviewers. These have been incorporated into the manuscript. These clarifications have significantly enhanced the readability and impact of the article. For the aforementioned reasons, I recommend its publication in Nature Communication.

Reviewer #3

(Remarks to the Author)

The authors have adequately addressed concerns from my previous review; I recommend publication as is.

April 5, 2025

Ref: our manuscript NCOMMS-25-02109-T

We are very grateful for the insightful and constructive comments of the Reviewers on our manuscript. Their remarks have helped us improve the quality of our work significantly. We have reproduced their comments verbatim and addressed every one of them, as detailed below. The revised version contain new experimental, computational and modelling data plotted in new Supplementary Figs. 2, 4 and 6.

We hope that our manuscript is now suitable for publication and look forward to hearing from you soon.

Sincerely,

Aloyse Degiron, on behalf of the authors.

Reviewer #1 (Remarks to the Author):

In this work the authors create a phase structured output from an incoherent source by careful design to allow electrical excitation in a small central region. Most of the figures show direct data pertaining to the device and its design, while the optical data is somewhat less convincing. The actual phase structure is shown in only one plot, Fig 2d, and then repeated later when the device is made more reliable. This is necessarily an indirect reconstruction, but that too makes the single result less appealing. Overall while the paper was easy to read, I feel that the motivation for the work is poor - why do we even need an LED emitting vortex beam when we have so many coherent sources of OAM? This critical question is never answered. What would I do with this LED device? Never mentioned. Mention of the record breaking 400 nm is said many times, but never why this is useful or desired. Broadband OAM control from white light sources has been shown in MANY paper, none of them cited. In fact, the introduction is not framed at all in terms of OAM/vortex sources, nor incoherent/broadband OAM control, with none of the literature in these two directions covered (only a modest mention in passing of a couple of papers that appear to have been selected at random). Instead it is centered on LEDs and the advances in this field. Without context or motivation, the work then comes across as a technical advance in device fabrication, with plausible confirmation. Here too though, the authors need to appreciate that there is a difference between OAM and phase singularities (see many papers on this by Michael Berry), so a swirling phase does not imply OAM. One would like to see the entire complex field to be sure we have a mode here, and for that matter, a mode of what type? I suggest that this be transferred to a journal that is read by a more specialised audience interested in solid state devices.

We thank the Reviewer for his/her thorough assessment of our manuscript but are sorry to discover that we were not able to convey our key points properly. **The main goal of this paper is to show how to endow LEDs with advanced non-trivial properties.** There is a huge competition worldwide in this domain, with studies published at a frantic pace on LEDs that do not emit purely unpolarized and isotropic emission. **This is a different endeavor than shaping incoherent emission with elements placed outside the source itself**, which is probably the literature that the Reviewer has in mind when he/she states that “broadband OAM control from white light sources has been shown in MANY papers” and that we should have cited and commented indeed.

The reasons for this interest in developing such LEDs are both fundamental and applied. On the fundamental side, the challenge is to gain total control over spontaneous emission (this is a dream shared by several communities beyond the LED community, including for fluorescence, phosphorescence and thermal emission, with the additional difficulty for LEDs that one must accommodate the electrical injection scheme). On the applied side, the interest is to develop LEDs capable of emitting functional/structured beams without the need of additional external elements that typically operate by filtering and rejecting a significant part of the signal, which is desirable for lowering energy consumption, OEM integration as well as in applications where lasers pose safety issues.

Our work represents a clear breakthrough because existing studies have so far introduced LEDs emitting directional and sometimes linearly-polarized light. Here, we demonstrate advanced phase manipulation with LEDs emitting optical vortices. This is not a small feat because the devices must not only accommodate advanced photonic management but also the charge injection scheme of a LED. To appreciate just how hard it is to make the photonic environment of a LED compatible with the charge injection scheme, we emphasize that it is a problem even for plain LEDs emitting non-directional and unpolarized radiation because a substantial part of the electromagnetic is lost by total internal reflection within the source.

We fully agree with the Reviewer that our introduction focuses on LEDs – it does so precisely because we address a key challenge for LEDs: the quest for LEDs capable of emitting beams with complex structure (and beyond LEDs, the quest toward full control of spontaneous emission in fluorescence, thermal emission, phosphorescence...). We explain in the introduction that the field is much less mature than for lasers, for which there has been several convincing demonstrations of integrated laser sources emitting non-trivial beams. In this regard, the papers that we cite for optical vortices were not chosen “at random”: they represent the state-of-the art in terms of vortex emission by integrated laser sources. We also cite the demonstrations made in fluorescence and thermal emission in order to explain that these architectures are not compatible with electrical pumping.

We also agree that we have not directly shown the effect of torque typical of OAM. In the revised version, we have removed the few instances where we mention OAM (although we are not aware of a single counter-example of a beam with a swirling phase between 0 and 2π around a single central singularity that does not carry OAM, including in Prof. Michael Berry’s works).

There is one remark from the Reviewer which we respectfully disagree with – the lack of convincing proof of vortex emission. We present original interference measurements that clearly show the phase singularity, which is the gold standard for experimental work on optical vortices. In addition, our experiments are backed by a model that makes a vortex interfere with a directional

beam with planar wavefronts (Fig. 2e). We realize, however, that we could elaborate more on the contribution of each wavelength to the interference images. We present additional experimental and modelling data in new Supplementary Fig. 6 on this point.

Changes made to address Reviewer's comment:

1/ We have sharpened the end of our introduction along the points outlined here:

- Page 3, we make it more explicit that references 26-31 specifically concern the state-of-the art in terms of vortex emission by integrated sources:

Several studies have demonstrated laser cavities emitting beams with inhomogeneous phase and polarization profiles such as optical vortices, vector beams and self-healing Airy beams²⁶⁻³¹. These demonstrations are typically performed with optically-pumped cavities rather than electrically-pumped ones.

- Page 4, we explicitly state that the main advance reported by this paper is to control the spontaneous emission directly within LEDs and that this is not the same thing as shaping the emission of an incoherent source using elements placed outside the source:

While broadband vortices can also be synthesized with optical elements placed outside conventional LEDs and other non-lasing sources (using spatial light modulators, spiral phase plates, volume phase holograms, uniaxial crystals, often combined with polarizing and/or spatial filtering elements)³⁹⁻⁴², this study shows a path forward for advanced beam structuration by directly acting upon the spontaneous emission process within the sources, facilitating their integration into more complex systems and offering potential alternatives to structured lasing light in applications where ocular safety and warmer colors are needed.

- In this new paragraph, we have cited the literature on white light vortices created by shaping the emission with elements placed outside the source itself (new refs 39-42).

2/ We have removed the couple of instances where we mention OAM when discussing our results.

3/ We have added additional experimental and computational data in Supplementary Fig. 6 to analyze the interference patterns.

Reviewer #2 (Remarks to the Author):

Title: Quantum Dot LEDs Emitting Broadband Vortex Beams

Authors: G. Boulliard, I. Roland, D. Schanne, M. Petolat, P. Filloux, E. Lhuillier, A. Degiron

The authors report on an electrically pumped optical source that emits a directional broadband vortex beam upon electrical injection. The approach consists of a metallic diffractive element that is disposed beneath an active layer region of embedded colloidal quantum dots. An aperture in the interposed dielectric spacer allows local injection of carriers, thereby confining the

emission to a micro-sized region at the centre of the diffractive element. The dispersion relation is retrieved by means of a spectrally-resolved Fourier imaging system and compared to the computed reflectivity of a simplified structure (a 2D grating) which approximates the system. The two are found to be in very convincing agreement. Then, in order to evidence a phase singularity over the entire range of wavelengths, a powerful method is employed. It involves the superimposition of four distinct gratings, each possessing specific phase at origin, on separate devices. The four resulting emission patterns are then combined to reveal the singularity. A vortex beam with unity topological charge is demonstrated. The weakness of the device (and its robustness in the absence of the metallic grating) when current passes through the device, leads the authors to consider a silicon diffractive element instead of metallic one. The same experimental procedure is pursued on this new architecture. Despite noisier measurements, the occurrence of singularity is once again observed, whilst robustness improved.

The article is well structured and the approach relevant. The scientific approach is rigorous. It is well written and informative.

I suggest that greater emphasis be placed on the practical implications for applications of the achievements reported here. The article would be significantly enhanced in terms of its impact and the emphasis of the results would be strengthened if the authors provided a more detailed explanation of how the emission of a vortex by an incoherent source with restricted spatial coherence represents a major advance.

In light of the aforementioned considerations, it is my recommendation that this paper be considered for publication in Nature Communications, pending the implementation of any requested corrections by the reviewers.

We would like to thank the Reviewer for his/her positive remarks and advice on further commenting the significance of our results. We have modified the last part of the introduction (with general considerations) and the conclusion (with more specific considerations) to address this point.

Questions/Comments:

To my opinion, the statement "hologram" appears somewhat exaggerated in the context of a diffractive element that, in essence, constitutes a simple canonical geometry structure: namely, a spiral.

Yes, we agree. We used the term "hologram" because this is how gratings with forked singularities and related geometries are named in textbooks and landmark reviews on optical vortices such as [Adv. in Opt. and Photon. 3, 161–204 (2011)]. Moreover, although the spiral can be constructed from a mathematical formula, it is technically easier for us to justify the choice of this geometry and its dimensions using basic holography arguments outlined in the Methods section and Eq. (1).

This being said, we accept the criticism—following your remark, we have noticed that we have used the term "hologram" 34 times in the original version of the manuscript and we agree that it puts way more emphasis than it should on this term.

Changes made to address Reviewer's comment:

We have removed the term "hologram" in 28 out of 34 instances. We have conserved the term "hologram" in introductory and concluding remarks as well as in the Methods section where we use basic holographic principles to design the spiral.

In the initial configuration, authors emphasise that no metallic pattern is introduced at the very center of the device to allow effective electrical injection. The rationale behind the exclusion of metal, given its potential to facilitate the transport of carriers, is not immediately apparent.

We fully agree that more information should be given.

Changes made to address Reviewer's comment:

- Page 6, we have removed the original remark: *note that we remove the very central part of the Au spiral to allow proper electrical injection*

- We have added the following text at the end of the same paragraph (bottom of page 6 and top of page 7): *It should be noted, in particular, that the spiral is slightly truncated so as to leave a 1.5 μm wide circular area without Au at the very center of the structure. Without this truncation, many n-doped QDs within the central PMMA hole would have been in contact with the Au spiral rather than with the electron transport layer, resulting in heterogeneous injection conditions that would have degraded the quality of the light emission.*

The density of n-doped quantum dots in the active zone is not specified. How many active QDs are involved when the pumping zone is restricted to the 3- μm size aperture in the insulator?

Thank you for pointing out that this information was missing. The size of the emissive QDs is ~ 4.5 nm and the interdot spacing after ligand exchange is about 1 nm, corresponding to a density of 6×10^6 QD/ μm^3 . In terms of the number of active QDs involved, the thickness of our layers is typically 5 monolayers. If we suppose that these monolayers uniformly coat the bottom of the 3 μm size aperture, the number of active QDs is approximately 10^6 . In reality, the QDs are not uniformly pumped because the different layers of the stack accumulate close to the PMMA walls (see new supplementary Fig. 2), reducing the effective area where the QDs are efficiently pumped. Supplementary Fig. 2 is partially reproduced here to document how the different layers of the stack behave in the vicinity of the central hole:

Changes made to address Reviewer's comment:

- We have added the information regarding the size, interdot spacing and density of the active QDs at the top of page S7 of the Supplementary information.
- We have added and commented scanning electron micrographs showing a cross-section of an actual device in new Supplementary Fig. 2.

A PMMA spacer is inserted between the active region and the diffractive element. What considerations led to the decision regarding the layer thickness?

Thank you for pointing out that this information was missing.

Changes made to address Reviewer's comment:

We have added the following sentence at the end of the paragraph addressing the fabrication of the PMMA layer in section 3.1 of the Supplementary information:

The PMMA thickness is a result of an empirical tradeoff: thinner layers were more sensitive to electrical leaks/shorts while thicker layers were detrimental to the next fabrication steps, the lithographed hole in the PMMA becoming too deep to ensure proper physical and electrical continuity between the ITO inside this hole and the rest of the ITO electrode.

As illustrated in Figure 3-e, the authors present the experimental and computed dispersion relations for Silicon-based sources. In this case, what is the coherence length?

Thank you for pointing out that this information was missing.

Changes made to address Reviewer's comment:

Page 15, we have added the following information to the sentence “Unlike LEDs operating with Au spirals (Fig. 1d), only one dispersive branch is visible in the dispersion map”:

Unlike LEDs operating with Au spirals (Fig. 1d), only one dispersive branch is visible in the dispersion map, with a linewidth corresponding to a spatial coherence length $\lambda/\Delta\theta$ of approximately 25 μm .

The reasons for the significantly higher noise levels observed with silicon spirals compared to gold ones remain ambiguous. Is this discrepancy merely attributable to the lower index of refraction of silicon, or do other factors play a more substantial role?

We have tried to explain the reasons in the last paragraph before the conclusion. Obviously, your question proves that our explanation was confusing and must be rewritten.

The index of refraction of silicon is not the culprit. In Supplementary Figure 8 (former Suppl. Fig. 5), we have measured and simulated a sample without Al electrode. All the other layers are the same as those of the LED operating with the a-Si spiral presented in Figure 3. Supplementary

Fig. 8 shows that the photoluminescence of the sample without Al electrode is coupled to a perfectly well-defined mode, indicating that a-Si is a great material for shaping the luminescence. Moreover, the signal-over-noise ratio is the best of all the structures presented in this study, as can be appreciated in a quantitative way with the cross-section of the dispersion relation displayed in the bottom panel of Figure 3f.

In other words, it is the presence of the Al electrode that severely degrades the signal over noise ratio. The reason can be understood by examining the field pattern of the mode without Al electrode. Supplementary Figure 8b shows that this mode extends well into the transparent substrate, implying that it is deeply frustrated when an Al electrode is inserted between the transparent substrate and the TiO₂ layer. The modified field distribution degrades the interactions with the QDs as well as the outcoupling of the guided modes into free space.

Changes made to address Reviewer's comment:

- We have rewritten the last paragraph before the conclusion to clarify the cause of the isotropic background:

To understand the origin of this background, it is useful to study the photoluminescence of a similar stack, but without the bottom Al cathode. As shown with the experimental dispersion relation of Supplementary Fig. 8a, almost all the photoluminescence is distributed in two well-defined crossing branches, with minimal signal outside these branches. In fact, the level of isotropic noise is the lowest of all the samples presented in this study, as shown in the bottom panel of Fig. 3f representing a cross-section of this dispersion relation. These measurements imply that the isotropic background of the LED operating with an a-Si spiral is caused by the Al electrode. Additional results, presented in Supplementary Fig. 8b, explain this behavior by the fact that the guided modes of a stack without a bottom Al cathode have a significant field extension within the glass substrate beneath the a-Si spiral. When an Al cathode is present, the field cannot extend into the substrate anymore, altering the interactions with the QDs as well as the outcoupling of the guided modes into free space. Nevertheless, these results demonstrate the potential of a-Si photonic structures for molding the light emission in very complex ways (in this regard, it is also worth noting that the photoluminescence experiments of Supplementary Fig. 8 show that the device without Al cathode generate a vector beam with an azimuthal polarization texture).

- We have added labels in Supplementary Fig. 8b to clarify that no Al electrode is present in this simulation.

Minor comments:

Some of the acronyms are not defined. For instance ETL. I guess this is electron transport layer. Same for HTL and NC.

Yes, you are right, ETL means electron transport layer. This acronym and the acronym HTL are defined in the caption of Figure 1. We forgot, however, to define NC in the original manuscript. Thank you for pointing out this omission.

Changes made to address Reviewer's comment:

We have defined the acronym NC in the caption of Figure 1, in the same sentence where we define ETL and HTL.

Reviewer #3 (Remarks to the Author):

The authors claim to demonstrate electrically-pumped LEDs that emit vortex beams. I think the central result is of high significance and the paper combines interesting and clever fabrication, measurement, and theory/simulation. I recommend the paper for publication in Nature Communications. My main reservation are certain areas of analysis/description that I think are some combination of oversold or not clearly explained. I am happy to classify these as "minor revisions" the main structure and presentation of the paper is solid and there is no need for any "major" change.

We would like to thank the reviewer for his/her positive appreciation of our work.

1. In the introductory section (page 3), the authors introduce some references on optically pumped beams and then state "However, applying these ideas to LEDs is a non-trivial matter, as the structures demonstrated up to date are not compatible with electrical pumping." I think the first part of this sentence is certainly correct; the second part however misses work on electroluminescent devices.

Namely authors show unidirectional beams within electroluminescent devices in:
- "Metasurface electrode light emitting diodes with planar light control" (<https://doi.org/10.1038/s41598-017-15254-3>) and
- "Metasurface Light-Emitting Diodes with Directional and Focused Emission" (<https://doi.org/10.1021/acs.nanolett.3c03272>)

Yes, we fully agree, thank you for bringing these two references to our attention. These papers report on LEDs emitting directional and/or linearly polarized beams. We have therefore added them in the paragraphs where we address the literature on this subject, on page 2 and page 3.

Changes made to address Reviewer's comment:

- We have added the two references in the manuscript (new references 18 and 19) and discuss these two references in two places in the text: in the bottom paragraph of page 2 and the first paragraph of page 3.

2. On page 8 the authors write that "These plasmonic contributions, involving metals with...explain why this TM mode manifests..." I don't think this is so evident from plotting the H-field intensity of the TM mode. I understand that they plot the H-field because it is characteristic of the mode, but it is the E-field intensity that determines the relative coupling to the emitter vs. the plasmonic metals. It would be nice to see this mode profile instead, or added to the supplement. In part, I wonder if the issue is more that the the TM mode couples weakly to the emitter rather than metal losses.

Yes, we fully agree that it is the E field that determines the coupling between the emitters and the guided mode and that the E-field should be plotted. The sentence “These plasmonic contributions...” erroneously implied that the material losses were the sole factor behind the lower intensity of the branch. We fully agree that the coupling efficiency also plays a key role in determining the intensity of the branch.

Since the main purpose of this paragraph was to discuss the branch width and the corresponding coherence length, we propose, for the sake of clarity, to remove any reference to the intensity of the branch in order to solely focus on its width. The latter does not depend on the coupling efficiency.

Changes made to address Reviewer’s comment:

- We have added the E-field distribution of the mode as a new Supplementary Fig. 4 and mentioned this new figure in the sentence before “These plasmonic contributions...” (page 8).
- We have corrected the erroneous sentence “These plasmonic contributions...” so as to discuss only the width of branch 2.

3. On page 10, the authors write that “This pattern unambiguously demonstrate that the LED...is generating a broadband vortex beam with a topological charge $m=1$...demonstrating that the phase singularity is encoded across the entire EL spectrum”. I think they overstate their case here, and elide issues with their results that are worth discussing.

- It is my understanding that an “ideal” or “unambiguous” vortex beam with charge $m=1$ would show the 2π azimuthal phase variation at all radial distances in Fig. 2d. Instead this is only really seen in the center, and the radial distance over which the phase evolution is clearly visible is noticeably smaller than in Fig. 3e. I think it would be better if the paper was clearer and more honest about the nature of the results so far. Namely they are probably seeing some vortex beam emission superimposed on non-trivial background emission that is not strongly mediated by the spiral structures.

Yes, we fully agree that there is non-negligible background emission. We tried to address this point in the original version of the manuscript with the help of Figure 3f. This figure displays cross-sections of the different dispersion relations shown elsewhere in the study. The cross-sections clearly show an isotropic background that is highest for the LED operating with the a-Si spiral.

Changes made to address Reviewer’s comment:

- We start discussing the issue of background emission much earlier than in the original version. Rather than waiting until Fig. 3f, we start discussing this issue with Fig. 1d with the following text on page 9:

Since the EL signal in Fig. 1d does not fall back to strictly zero outside the TE- and TM-like branches, this directional emission is accompanied by a fainter isotropic background corresponding to the fraction of the EL that is not coupled to the modes of interest.

- I further suspect that the degree to which this looks like an “ideal” or “unambiguous” vortex beam is probably significantly wavelength dependent. It is certainly possible that at some parts of the spectrum there is no vortex beam effects evident at all. That all wavelength photons are collected to produce Fig. 2d does not prove that all wavelengths have an emission characterized by the pattern in Fig. 2d. If the authors want to show this clearly, they could use three notch filters at ~ 1200 , 1400 , and 1600 to plot something like Fig. 2d at different portions of the emission spectrum. Absent a measurement like that, I think they should be less certain and more precise about the degree to which “the phase singularity is encoded across the entire EL spectrum”. Particularly in light of my previous bullet.

Thank you for pointing out that we were too vague in supporting the claim of broadband vortex emission. Experimentally, the signature of broadband vortex emission can be directly seen in the phase plot of Fig. 2d due to the fact that the modes of interest are emitted in a hollow cone with a wavelength-dependent aperture (Fig. 1d). Specifically, the wavelengths closest to the crossing of the TE-like branches 1a and 1b contribute to the signal at the center of the image and the other wavelengths contribute to the signal at increasingly longer distances from this center.

Moreover, we agree with the reviewer that the phase singularity is only clearly visible in the center area of Fig. 1d, and more precisely in a circle with a radius $k_{//}/k_0 \sim 0.3$. This radius corresponds to the point where all the experimental branches of the dispersion relation of Fig. 1d have disappeared in the isotropic background, at $\lambda \approx 1280$ nm. There is no reason that vortex emission does not occur at smaller wavelengths, but it is hidden in the isotropic background so we agree that our experiments do not support the claim of vortex emission for these wavelengths.

Following the Reviewer’s suggestion, we illustrate these points with measurements taken with three narrow bandpass filters. We solely analyze one of the four interferometric devices, since filtering the signal allows one to reveal the distinctive crescent shape of an annular beam with a phase singularity around which the phase rotates from $-\pi$ to $+\pi$ that interferes with an annular beam with planar wavefronts. We chose three bandpass filters for which the modes of interest evidenced in Fig. 1d are clearly emerging from the noise:

These results are plotted and analyzed in new Supplementary Fig. 6. Importantly, these new results are backed by our scalar model which makes a vortex interfere with a directional beam with planar wavefronts.

Changes made to address Reviewer's comment:

- We have rewritten pages 11 and 12 to better substantiate the claim of broadband emission. We explain that experimentally, we can only claim broadband emission in the wavelength range where the directional branches of Fig. 1d stand out from the noise (from 1280 nm to 1600 nm).
- As a corollary, even if the phase singularity is likely encoded across the entire EL spectrum, we now only claim broadband vortex emission from 1280 nm to ~1600 nm, since the background noise hid the vortex emission beyond this range.
- We have added measurements taken with three narrow bandpass filters in new Supplementary Fig. 6.

4. I did not understand the argument about how the Au spiral "seems to favor irreversible current leaks". I get it in the context of their device results, but I don't understand it in the context of how the device should work. Like, if I just made a planar (i.e. unpatterned) device where Au is blanket deposited atop the ETL, why would that cause "current leaks"? That whole bottom section is still the Cathode and should be electrically isolated from the anode except through current paths that pass through the semiconductor active area. My understanding is that we want to avoid direct metal contact to the semiconductor active area to eliminate recombination effects and to create selective contacts, not to prevent "current leaks". Is this just an issue with the quality of the PMMA film deposited atop Au spiral vs. when it's deposited atop a-Si? If I'm missing something, some clarifying cartoons and/or device schematics would be helpful.

We agree with you that more clarification is needed. By "irreversible current leak", we mean the same thing as what is happening when the gate oxide of a field effect transistor experiences an irreversible breakdown following a defect build-up. In our case, the sensibility of the PMMA layer to electrical breakdown can be explained by the fact that Au is not continuous atop the TiO₂ layer, creating voltage and current heterogeneities that can only favor defect build-ups and shorts through the PMMA layer.

Changes made to address Reviewer's comment:

- On page 12, we have reworded the sentence with "irreversible current leaks" which now reads: *"The relative fragility of the devices can be unambiguously traced to the Au spiral, which seem to induce heterogeneities in the applied electrical current and equipotential lines that are substantial enough to induce irreversible shorts through the PMMA layer."*

Other changes

- We corrected typos and wording.
- We extended the acknowledgment section.
- We complied with the editorial policy requests